# Does Melatonin Enhance Twin Lamb Survival in Commercial Merino Flocks in Australia?

**DOI:** 10.3390/ani15070946

**Published:** 2025-03-25

**Authors:** Alyce M. Lowe, David O. Kleemann, Jennifer M. Kelly, Andrew N. Thompson, Jarryd Krog, William H. E. J. van Wettere, Alice C. Weaver

**Affiliations:** 1Turretfield Research Centre, South Australian Research and Development Institute, Department of Primary Industries Regions South Australia, Livestock Innovation and Welfare, Rosedale, SA 5350, Australia; dave.kleemann@sa.gov.au (D.O.K.); jenkelly@internode.on.net (J.M.K.); alice.weaver@sa.gov.au (A.C.W.); 2Centre for Animal Production and Health, Food Futures Institute, Murdoch University, Murdoch, WA 6150, Australia; andrew.thompson@murdoch.edu.au (A.N.T.); jarryd.krog@murdoch.edu.au (J.K.); 3Davies Livestock Research Centre, School of Animal and Veterinary Sciences, The University of Adelaide, Adelaide, SA 5001, Australia; william.vanwettere@adelaide.edu.au

**Keywords:** melatonin, gestation, ewes, lamb survival, lamb growth, twin, Merino

## Abstract

High incidence of twin lamb mortality limits the reproductive efficiency of Merino sheep flocks. This study assessed if supplementing twin-bearing Merino ewes managed on commercial properties in Australia with melatonin implants during late pregnancy would increase lamb survival and weight at weaning. Implanting one 18 mg melatonin pellet (Regulin^®^, Ceva Animal Health Australia, Glenorie, New South Wales. Australia) did not improve either the lamb survival or the weight at weaning. This strategy to improve twin lamb survival cannot be recommended at this stage on commercial farms.

## 1. Introduction

Increasing neonatal survival is integral to improving the reproductive efficiency of sheep. In Merino flocks, the predominant breed in Australia, the high incidence of lamb moralities has long been a problem for the sheep industry, where the animals are grazed under extensive conditions. The average twin lamb survival is approximately 75%, with this high incidence of pre-weaning lamb mortalities being both an animal welfare and economic issue [1,2] Despite improvements in the genetics and management of ewes [3], improving twin lamb survival of Merinos has proven difficult due to the multifaceted impacts of nutrition, climate, predation, disease, genotype, and ewe/lamb behaviour.

A new approach to increase neonatal survival, advanced by Flinn, et al. [4], is to supplement the ewes during late gestation with melatonin, a hormone released from the pineal gland at night [5]. Previous research has shown that melatonin can ameliorate the effects of oxygen deprivation during the birth process through an improved maternal–fetal blood flow [6,7,8,9], increased nutrient supply [10], and fetal growth [11,12]. Melatonin diffuses through the placenta, protecting the fetal brain from the hypoxic-ischemic damage often caused by the prolonged parturition process and birth injury, commonly associated with twin lambs [13] Melatonin, administered either by a slow-release implant or via a feed supplement, can negate the negative impacts on the lamb behavioural abilities, such as standing soon after birth, contacting the udder, and following the dam [5,14,15]. Additionally, an improved blood flow and nutrient supply not only increased the fetal growth but also enhanced brown adipose tissue deposition, which is important for neonatal thermogenic activity [16,17]. There is also evidence that implanting Assaf ewes with melatonin during late gestation can improve milk production and increase the immunoglobulin concentration in colostrum, vital for survival post birth [18].

An intensive indoor study examined maternal melatonin treatment in single- and twin-bearing Merino ewes and reported a tendency for an improved survival in twins but not singles [19]. Contrary to the expectation, none of the physiological parameters, including the weight at birth to weaning, rectal temperature, lamb serum IgG concentration, meconium score, glucose concentration, nor the behavioural traits (latency to stand and suck, vigour score) were improved by melatonin treatment, with the exception that gestational length and the survival of the second-born versus first-born lambs were increased. The latter finding is important, as it supports the assertion that melatonin has neuroprotective properties which help the second-born escape the deleterious effects of prolonged parturition. These impacts on twin lamb survival were tested under controlled grazing conditions at the Minnipa Research Centre and confirmed that melatonin increased survival by 14% (80 vs. 94%); yet did not alter weight at birth or weaning [20]. As melatonin implants cost approximately $7.50 AUD plus labour per ewe, the verification of the impact of melatonin on lamb survival under a broad range of commercial conditions is required before the management practice can be recommended to sheep producers.

The current study examined if improvements in twin lamb survival, occurring due to the maternal melatonin treatment, would manifest in commercial Merino flocks managed under extensive grazing conditions in Australia. We hypothesised that the treatment of commercially grazed Merino ewes with melatonin during late pregnancy improves the growth and survival of their twin-born lambs.

## 2. Material and Methods

### 2.1. Animal Ethics, Property Locations, and Climate

This experiment was part of a large-scale, long-term collaboration project between The South Australian Research and Development Institute (PIRSA) and The University of Adelaide. Funding was provided by Meat and Livestock Australia Ltd (North Sydney, NSW. 2060) (Project: L.L.M.S. 0015). All procedures were approved by the University of Adelaide Animal Ethics Committee (Project number UofA #: 34902) and were conducted in accordance with the Australian Code for the Care and Use of Animals for Scientific Purposes, 8th edition.

Experiments were conducted on three commercial Merino enterprises located in the cereal-sheep zones of south (*n* = 2) and western Australia (*n* = 1). All locations experience a Mediterranean-type climate, with rainfall occurring predominantly during the winter. Specific details of the sites’ locations and climate are presented in Table 1.

### 2.2. Animal Management, Experimental Design, and Measurements

At each site, multiparous ewes (*n* = 1172 ewes between 3 and 6 years old) were naturally mated to Merino rams in December 2021 (Keith, South Australia, *n* = 400), January 2022 (Jamestown, South Australia, *n* = 400), and February 2022 (Frankland River, Western Australia, *n* = 372) for a mating period of 42, 48, and 21 days, respectively. During the allocation to treatments, ewe body condition score (BCS) was estimated according to the description by Russell et al. [21]. Across all three sites, 1171 twin-bearing Merino ewes with a mean BCS 3.2 ± 0.1 were randomly allocated to one of two treatment groups: no melatonin implant (control) or an 18 mg melatonin implant (Regulin^®^) inserted behind the ear, as recommended by the manufacturer. The melatonin treatment during gestation began later at Keith compared with the other two sites (Keith = 105 days after rams were exposed to ewes or day of gestation (dG), Jamestown = dG 92, and Franklin River = dG 89). Further details of the timing of treatment, the number of ewes allocated within treatment, and the replicate groups on each property and the BCS are presented in Table 2. The BCS was measured on the day of the melatonin implant administration during gestation on all properties, and at tail docking and weaning at Franklin River. Ewes on each site were identified with an individual electronic ear tag (EID) (Keith and Frankland River) or treatment specific coloured ear tag (Jamestown) and were treated as a single flock until about 6 to 8 weeks prior to the onset of lambing at Keith and Jamestown and 2 weeks at Franklin River, when they were separated into their treatment groups and replicates. The Keith property managed a single replicate of each treatment, while Jamestown and Franklin River managed two and three replicates, respectively. The lambing period was managed as per normal farm practice. The ewes and lambs were counted at tail docking and weaning (the latter at approximately 87 days of age), and the lambs were weighed at weaning. Ewes’ BCS and weight were recorded at tail docking and weaning at Franklin River only.

### 2.3. Statistical Analysis

A statistical analysis was conducted using IBM SPSS Statistics version 28 software. A Pearson’s chi-square test (χ^2^) was used to analyse the effects of melatonin treatment and property location on lamb survival at tail docking and weaning. A general linear model was used to determine the treatment differences on ewe BCS at selection and on lamb weaning weight. Bonferroni’s correction was used to determine the pairwise comparisons between the treatment groups and property location. Significance was accepted when the probability value was (*p*) ≤ 0.05. Survival data are presented as a cumulative data percentage (%) and total animal number (*n*), and ewe BCS and liveweight (LW) and lamb LW data are presented as least squares mean ± standard error of mean (SEM).

## 3. Results

### 3.1. Body Condition Score and Live Weight of Ewes

Ewe BCS did not differ between the melatonin and control treatment groups within the properties during gestation; however, there were significant differences between the properties (Table 2) after the allocation of treatments. Ewes at Keith had a higher BCS (3.4) compared to those at both Jamestown (3.3) and Frankland River (2.6).

At the Franklin River site there were no significant differences between the control and melatonin treatments for either BCS (Table 2) or LW at tail docking (62.8 ± 0.6 versus 62.3 ± 0.6 kg) or weaning (69.7 ± 0.7 versus 68.3 ± 0.7).

### 3.2. Lamb Survival

There were no significant treatment differences (*p* > 0.05) for lamb survival at either tail docking or weaning when compared either between or within sites (Table 3).

Fewer lambs survived to tail docking at Jamestown (73%) compared with those at either Keith (79%) or Frankland River (78%) (Table 3; *p* < 0.001). These between-property differences for survival at tail docking were reflected at weaning (72% vs. 77% and 75%, respectively).

### 3.3. Lamb Weight at Weaning

At all the sites, the twin lambs born to control ewes were 0.4 kg heavier (*p* = 0.020) at weaning compared to the twin lambs born to melatonin treated ewes (Table 4). This treatment difference occurred mainly due to the control lambs being 0.8 kg heavier (*p* < 0.001) at the Keith site, as the treatment differences at the other two sites were not significant (Table 4).

There was a difference in weaning weight between sites (*p* = 0.001; Table 4) with the weight of lambs at Keith being lighter compared with those at both Jamestown and Frankland River.

## 4. Discussion

The potential to improve the survival of twin-born lambs by treating Merino ewes with melatonin implants during mid-pregnancy was tested on commercial flocks managed under extensive conditions, as normally experienced in Australia. However, unlike the similar trials conducted under both intensive [19] and extensive [20] research conditions, there was no improvement in twin lamb survival in any of the three flocks investigated, indicating that the commercial adoption of the technology cannot be recommended at this stage.

As previously mentioned, the positive findings of maternal supplementation of melatonin reported by Flinn et al. [19,20] and the short report by Davis et al. [22] where the melatonin (Regulin^®^) was also implanted in Merino ewes during mid-pregnancy reported a 7% improvement (*p* < 0.05) in lamb survival to weaning (68% vs. 75%) in one year, but no improvement in another year; however, there was no reference to the type of birth. Whereas, in the two Flinn et al. [19,20] studies, melatonin was administered to twin-bearing ewes, resulting in a 13% (72.9% vs. 85.9%; *p* < 0.07) and 14.4% (79.6% vs. 94.0%: *p* < 0.002) increase in lamb survival, respectively. The smaller improvement in survival found by Davis, Green, and Abecia [22] may be due to the inclusion of single- and twin-bearing ewes in their experiment, where the proportion of single-born lambs was likely to have been greater than multiple-born lambs, and, subsequently, the response to melatonin was likely to be small (or non-existent) in single lambs, as demonstrated in Flinn et al. [19]. Nevertheless, an inconsistent response to melatonin for survival was observed between the years [22] and between the experiments (Flinn et al. [19,20] and the current study), which suggests that further studies are required to understand the underlying sources of variation and to enable better prediction of when melatonin may be beneficial.

Some of the underlying sources of variation that may influence the response in lamb survival to melatonin have been examined previously, including dose rate and method of supplementation, season, and type of birth. A dose of one versus two implants (18 mg vs. 36 mg) was examined by Flinn et al. [20] and indicated that one implant was as efficacious as two, with survival rates of 94% vs. 92.5%, respectively. Flinn et al. [19] also reported that feeding the ewes’ melatonin in a capsule in the afternoon to extend the rise in endogenous melatonin secreted at night was as effective as the delivery via a single melatonin implant, with lamb survival rates of 85.5% vs. 85.9%, respectively. In the same experiment, the season of supplementation (autumn, with short days and high endogenous melatonin, versus spring, with long days and low endogenous melatonin) was investigated as a possible source of variation. The positive responses to exogenous melatonin, either administered through a single implant or supplemented daily via a capsule, were independent of the seasonal changes in endogenous melatonin. Flinn et al. [19] also determined that the type of birth was important, as there was little or no response in single-born lambs with the main response in survival seen in second-born lambs of the twin cohort. This result led Flinn et al. [19] to conclude that melatonin was especially important for the lambs exposed to prolonged births, with the hormone providing neuroprotective properties during and after the birth process. Other factors worthy of examination are the nutritional status and age of the ewe, breed, and timing of supplementation. Experiments by Flinn et al. [19] were conducted, where the ewes were fed to meet nutritional requirements; however, it is possible that the ewes fed at lower levels may divert nutrients differentially and alter the ability of melatonin to impact fetal growth, thermogenesis, and colostrum production. Melatonin response might also be regulated by the age of the ewe, particularly where young, growing ewes have a high demand for protein required for their own body growth as well as for fetal development and mammogenesis. However, a recent study by Haslin, et al. [23] in the Merino, where the ewes were lambed at one year of age, does not support this supposition; birth weight, growth rate, and lamb survival were not increased by the melatonin treatment. In addition, it is likely that breeds of higher fertility than the Merino, particularly in those breeds where the incidence of triplet-bearing and higher order births is prevalent, will benefit from the ability of the melatonin treatment to provide neuroprotection from the effects of a prolonged parturition. Finally, delaying and shortening the period of melatonin supplementation relative to parturition warrants investigation since we suggested that improvements in lamb survival were primarily induced by neuroprotection [20]. It is not known if a chronic or acute hypoxia occurs during gestation in sheep and whether the melatonin treatment in the weeks prior to parturition is beneficial. As the single-born lambs from ewes treated with melatonin from dG 80 did not respond in either survival or birth weight, this would suggest that gestational hypoxia may not be an issue [19].

Lamb growth to weaning was reduced by the melatonin treatment at the Keith site (0.8 kg; *p* < 0.001), while the treatment had no influence on lamb LW at weaning at the other two sites. The latter result concurs with those of Flinn et al. [19,20] while Davis, Green and Abecia [22] reported an increase in weaning weight (1.3 kg; *p* < 0.01) in the first year, but a non-significant decrease (0.9 kg) in the second year. While the improvement in growth to weaning reported by Davis, Green and Abecia [22] could be a reflection of both an enhanced fetal growth [4] and an improved milk composition (Canto et al. [18] due to the effects of melatonin on blood flow and nutrient supply, the lack of response in birth weight and colostrum intake in the studies of Flinn et al. [19,20] and a negative response in weaning weight in the current study would suggest otherwise.

## 5. Conclusions

Despite increases in the twin-lamb survival under both intensive and extensive research conditions, the supplementation of melatonin in twin-bearing Merino ewes managed on three commercial properties in Australia did not improve either the lamb survival or weaning weight. Further studies are required to ascertain why the response of these parameters to melatonin supplementation is so variable before the technology can be adopted commercially.

## Figures and Tables

**Table 1 animals-15-00946-t001:** Location and climate information for the three experimental sites. South Australia (SA) and western Australia (WA).

Site	Keith (SA)	Jamestown (SA)	Frankland River (WA)
Location	−36.231435° S, 140.409961° E	−33.078486° S, 138.59832° E	−34.196019° S, 117.106824° E
Annual rainfall, mm			
2022	562	553	475
Mean	460	440	590
Annual mean temperature, °C			
2022	21.8	21.2	20.3
Mean	22.3	21.9	20.9

**Table 2 animals-15-00946-t002:** Body condition score (BCS) and the number of ewes (*n*) for each treatment (control, melatonin (Regulin^®^ implant)), selected at each of the commercial properties and within each property. BCS is presented as mean ± SEM.

Property	Control		Melatonin		
	Ewes (*n*)	BCS	Ewes (*n*)	BCS	*p* Value
**All ewes**	585	3.2 ± 0.1	586	3.2 ± 0.1	0.525
**Between properties**					**0.001**
**Within properties**					0.162
*Keith* ^a^					
Gestation	200	3.4 ± 0.1	200	3.4 ± 0.1	
*Jamestown* ^b^					
*Gestation*	201	3.3 ± 0.1	199	3.3 ± 0.1	
*Frankland River* ^c^					
Gestation	186	2.6 ± 0.1	186	2.7 ± 0.1	
Tail docking	180	2.5± ± 0.1	178	2.6 ± 0.1	
Weaning	177	3.0 ± 0.1	176	3.0 ± 0.1	

^abc^ Superscripts indicate significant differences between properties (*p* = 0.001).

**Table 3 animals-15-00946-t003:** Survival of twin-born lambs from Merino ewes either administered a Regulin^®^ implant (melatonin) or no implant (control) during mid-late gestation.

Twin Lamb Survival (%)	Control	Melatonin	χ^2^	*p* Value
*Overall Sites*				
Ewes (*n*)	558	563		
Total lambs expected (*n*)	1116	1126		
Lambs marked/lambs expected, % (*n*)	77.4 (864)	75.7 (852)	0.959	0.327
Lambs marked/100 ewes	155	151		
Lambs weaned/lambs expected, % (*n*)	75.1 (838)	74.0 (833)	0.364	0.546
Lambs weaned/100 ewes	150	148		
*Data presented for lambs at tail docking (marking) and weaning with data combined for the three sites and between individual sites.*				
** *Individual sites* **				
*Keith*				
Ewes (*n*)	195	196		
Total lambs expected (*n*)	390	392		
Lambs marked/lambs expected, % (*n*)	79.7 (311)	78.1 (306)	0.332	0.564
Lambs marked/100 ewes	159	156		
Lambs weaned/lambs expected, % (*n*)	79.0 (308)	76.0 (298)	0.978	0.323
Lambs weaned/100 ewes	158	152		
*Jamestown*				
Ewes (*n*)	192	194		
Total lambs expected (*n*)	384	388		
Lambs marked/lambs expected, % (*n*)	74.7 (287)	71.6 (278)	0.939	0.333
Lambs marked/100 ewes	149	143		
Lambs weaned/lambs expected, % (*n*)	72.1 (277)	70.9 (275)	0.150	0.698
Lambs weaned/100 ewes	144	142		
*Franklin River*				
Ewes (*n*)	171	173		
Total lambs expected (*n*)	342	346		
Lambs marked/lambs expected, % (*n*)	77.8 (266)	77.5 (268)	0.010	0.920
Lambs marked/100 ewes	156	155		
Lambs weaned/lambs expected, % (*n*)	74.0 (253)	75.1 (260)	0.124	0.725
Lambs weaned/100 ewes	148	150		

**Table 4 animals-15-00946-t004:** Weaning weight (kg) of twin lambs born to ewes which either received melatonin (Regulin^®^) implant or no implant (control) during mid-gestation. Data are expressed as mean ± SEM.

Variable/Property	Control	Melatonin	*Weaning Age, d*		*p* Value	
			Median	Range	Treatment	Location
Overall weaning weight, kg	27.2 ± 0.1	26.8 ± 0.1	87	43–116	**0.020**	**0.001**
*Keith*						
Weaning weight, kg	24.5 ± 0.2	23.7 ± 0.2	69	48–90	**0.001**	
Property average, kg	24.1 ± 0.1 ^a^					
*Jamestown*						
Weaning weight kg	27.2 ± 0.2	26.9 ± 0.2	87	63–111	0.496	
Property average, kg	27.1 ± 0.2 ^b^					
*Frankland River*						
Weaning weight, kg	29.1 ± 0.2	29.7 ± 0.2	106	95–116	0.643	
Property average, kg	29.8 ± 0.1 ^c^					

^abc^ Superscripts indicate significant differences between properties (*p* = 0.001).

## Data Availability

No new data were created or analyzed in this study. Data sharing is not applicable to this article.

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
