# Peer review of "Does Melatonin Enhance Twin Lamb Survival in Commercial Merino Flocks in Australia?"

_animals, 2025, doi:10.3390/ani15070946_

Round 1

Reviewer 1 Report

Comments and Suggestions for Authors

Dear authors,

I will recommend your article for publication, but I have several questions about the research results. They are short, but very serious to be asked.

L102 – Add body parameters was investigated for BCS calculation.

L121 – Why you are not investigating level of melatonin in blood serum during experiment? How you proofing what melatonin implant significantly changing level of hormone in serum? Why you don’t establish placebo implant to control group? What is wrong with not survived lambs? Can melatonin theoretically prevent these cases?

L133 – “Ewe BCS did not differ between the Melatonin and Control treatment groups within properties during gestation” – As I understand, you were selected for research animals with BCD around 3.1. For what this conclusion?

L140 – Not clear – what is three P Value in last column? Is it not connected to p-value in table footnote?

L167 – You were comparing Property average of different places in different Weaning age and were found differences. Is it true conclusion?

Open Review

Quality of English Language

( ) The English could be improved to more clearly express the research.
(x) The English is fine and does not require any improvement.

Yes

Can be improved

Must be improved

Not applicable

Does the introduction provide sufficient background and include all relevant references?

( )

(x)

( )

( )

Is the research design appropriate?

( )

(x)

( )

( )

Are the methods adequately described?

( )

(x)

( )

( )

Are the results clearly presented?

( )

(x)

( )

( )

Are the conclusions supported by the results?

( )

(x)

( )

( )

Comments and Suggestions for Authors

Dear authors,

I will recommend your article for publication, but I have several questions about the research results. They are short, but very serious to be asked.

Comment 1: L102 – Add body parameters was investigated for BCS calculation.

Response 1: The technique of measuring BCS involves a subjective “feel” assessment of fat cover over the lumbar region of the sheep’s back.  The technique is adequately described by Russell et al (1969) without the need for further description in this paper.  The reference is given on line 103 in Material and Methods but is missing from the list of References.  It has now been added at line 313.

Comment 2 L121 – Why you are not investigating level of melatonin in blood serum during experiment? How you proofing what melatonin implant significantly changing level of hormone in serum? Why you don’t establish placebo implant to control group? What is wrong with not survived lambs? Can melatonin theoretically prevent these cases?

Response 2: The main aim of this experiment was to investigate if supplementation of melatonin to ewes during late pregnancy improves the survival of lambs and their growth to weaning.  Our previous studies (Flinn et al., 2020b,c) established that supplementation does improve lamb survival and that the effect is manifest during the parturition process where the survival of second born twin lambs is enhanced (Flinn et al., 2020b) and is likely to be due to melatonin’s relief of hypoxia during and after the parturition process.  In addition, we have reported the effects of both implants and orally administered melatonin on serum/plasma levels in the ewe (Swinbourne et al (202?); a sentence has been added in the Materials and Methods at line 106 after “manufacturer” describing the melatonin levels achieved when using implants, namely:

“We have previously established that plasma melatonin levels are elevated by administration of implants; for example, levels were raised from about 200 pg/ml to 600 pg/ml for control and melatonin treated ewes when implanted at day 80 and when measured at 21h (Lowe et al., in preparation).”

 It is unlikely that there is a placebo effect due to the implant as this was established in multiple studies when the melatonin implants were first developed during 1980-1990 era (for example Kennaway and Gilmore 1985  J. Reprod Fertil 73, 85-91; Williams et al 1992 Anim Reprod Sci 30, 225-258).

Comment 3: L133 – “Ewe BCS did not differ between the Melatonin and Control treatment groups within properties during gestation” – As I understand, you were selected for research animals with BCD around 3.1. For what this conclusion?

Response 3: We can conclude from this sentence that the allocation process was successful in selecting ewes at random to establish populations groups of similar mean BCS and variance within properties. Therefore we have added “after allocation of treatment to line 137.

Comment 4 L140 – Not clear – what is three P Value in last column? Is it not connected to p-value in table footnote?

Response 4: The P-value corresponds with the differences in the first three categorises listed in the table; all ewes P = 0.525, between properties p = 0.001, within properties P = 0.162. As properties is the only significant effects, we have added the superchips a, b and c, beside the property name, which has been supported by the superscript provided below the table.  

Comment 5: L167 – You were comparing Property average of different places in different Weaning age and were found differences. Is it true conclusion?

Response 5: We think that the overall mean for the treatment groups is a valid comparison even though the weaning ages differed between sites and since the age within sites were the same.  We go on to explain in the second sentence (Line 159) that the overall treatment difference was mainly due to a within site difference at Keith, giving the reader a full explanation of why the overall mean was significantly different at P < 0.02.

Regards,

Submission Date

25 February 2025

Date of this review

28 Feb 2025 06:37:07

Regards,

Reviewer 2 Report

Comments and Suggestions for Authors

Line 20-21 – consider clarifying the context i.e., was this for “research flocks” (whereas your study is now considering commercial flocks; might make it clearer for a reader)

Line 33 – there’s a random “introduction” in the keywords – guessing the formatting had an issue

I found the introduction a pleasure to read – introduced the topic in a very logical, thoughtful way

Section 2.2 – it would be nice to know how many ewes (treatment & control) were on each of the 3 sites at this stage rather than having to read ahead to the results tables.

Section 2.2 – I am I correct in assuming that the only matching was on bearing status – I’m not saying I disagree with this, but in the abstract, it alluded to matched controls (“corresponding controls” – line 25) whereas this describes random allocation to txt vs control. Consider changing this wording to avoid any confusion.

Line 119 – missing a “t” from weight

Results – what was the ewe mortality in this study, overall, and across the sites/groups?

Section 3.3. – something odd going on (Error! Reference source not found appearing in text)

Discussion – what was the cost of the implant per ewe? Useful context for a reader.  

Question for my own interest – did you consider necropsies on the lambs to look at likely cause of death between the groups/sites? I appreciate how difficult this is in commercial situations but would be interesting data to add to this overall picture. 

Author Response

Open Review

Quality of English Language

( ) The English could be improved to more clearly express the research.
(x) The English is fine and does not require any improvement.

Yes

Can be improved

Must be improved

Not applicable

Does the introduction provide sufficient background and include all relevant references?

(x)

( )

( )

( )

Is the research design appropriate?

(x)

( )

( )

( )

Are the methods adequately described?

( )

(x)

( )

( )

Are the results clearly presented?

( )

(x)

( )

( )

Are the conclusions supported by the results?

(x)

( )

( )

( )

Comments and Suggestions for Authors

Comment 1: Line 20-21 – consider clarifying the context i.e., was this for “research flocks” (whereas your study is now considering commercial flocks; might make it clearer for a reader)

Response 1: In line 20 we have inserted “in research flocks” after “shown”.

Comment 2: Line 33 – there’s a random “introduction” in the keywords – guessing the formatting had an issue

Response 2: Yes, This has been removed from the key words

Comment 3: I found the introduction a pleasure to read – introduced the topic in a very logical, thoughtful way

Response 3: Thank you for the feedback. It is greatly appreciated.

Comment 4: Section 2.2 – it would be nice to know how many ewes (treatment & control) were on each of the 3 sites at this stage rather than having to read ahead to the results tables.

Response 4: L100-101, we have inserted the number of ewes at each site (Keith, South Australia. n = 400), January 2022 (Jamestown, South Australia. n = 400), and February 2022 (Frankland River, Western Australia. n = 372)

Comment 5: Section 2.2 – I am I correct in assuming that the only matching was on bearing status – I’m not saying I disagree with this, but in the abstract, it alluded to matched controls (“corresponding controls” – line 25) whereas this describes random allocation to txt vs control. Consider changing this wording to avoid any confusion.

Response 5: In line 25 we have deleted “corresponding” to allay any confusion”

Comment 6: Line 119 – missing a “t” from weight

Response 6: “Weight” has been corrected. L121

Comment 7: Results – what was the ewe mortality in this study, overall, and across the sites/groups?

Response 7: Ewe mortality data has been provided in Table 3 with the number of ewes provided in under each property. We have also added the total number of ewes selected for this experiment. L 99 At each site, multiparous ewes (n = 1172 ewes between 3 – 6-year-old)  

Comment 8: Section 3.3. – something odd going on (Error! Reference source not found appearing in text)

Response 8: Fixed or to be fixed by Editor

Comment 9: Discussion – what was the cost of the implant per ewe? Useful context for a reader. 

Response 9: Line 74-75 now reads “As melatonin implants cost approximately $7.50 AUD plus labour, the verification of the impact ..”

Question for my own interest – did you consider necropsies on the lambs to look at likely cause of death between the groups/sites? I appreciate how difficult this is in commercial situations but would be interesting data to add to this overall picture. 

No – data was not collected – very difficult to do and obtain meaningful results on commercial properties

Reviewer 3 Report

Comments and Suggestions for Authors

In this study, Lowe et al. examined whether treatment of Merino ewes with melatonin improved twin survival. This paper is well written, so I recommend it to be published in Animals after a minor revision of the manuscript:

  1. Do you have a hypothesis regarding the significant difference in BCS and weaning weight between the sites? If you have a hypothesis, it would be good if you could discuss it.
  2. Some examples of typing errors can be found:

Line 33: “Introduction” should be removed

Line 158: “(Error! Reference source not found.)” should be removed

Line 161: “(Error! Reference source not found.)” should be removed

Author Response

Open Review

Quality of English Language

( ) The English could be improved to more clearly express the research.
(x) The English is fine and does not require any improvement.

Yes

Can be improved

Must be improved

Not applicable

Does the introduction provide sufficient background and include all relevant references?

(x)

( )

( )

( )

Is the research design appropriate?

(x)

( )

( )

( )

Are the methods adequately described?

(x)

( )

( )

( )

Are the results clearly presented?

(x)

( )

( )

( )

Are the conclusions supported by the results?

( )

(x)

( )

( )

Comments and Suggestions for Authors

In this study, Lowe et al. examined whether treatment of Merino ewes with melatonin improved twin survival. This paper is well written, so I recommend it to be published in Animals after a minor revision of the manuscript:

Comment 1: Do you have a hypothesis regarding the significant difference in BCS and weaning weight between the sites? If you have a hypothesis, it would be good if you could discuss it.

Response 1: It is difficult to hypothesis because BCS of ewes at the Keith site where weaning weight was reduced by melatonin treatment was not measured.

Comments 2, 3 and 4. Some examples of typing errors can be found:

Response to comments 2, 3 and 4. Yes, there appears to be some formatting errors which the authors have also identified during proofing. That have all been fixed and highlighted for the editor to review.

Comment : Line 33: “Introduction” should be removed

Fixed

Line 158: “(Error! Reference source not found.)” should be removed

Fixed

Line 161: “(Error! Reference source not found.)” should be removed

Fixed

Reviewer 4 Report

Comments and Suggestions for Authors

To improve the efficiency of livestock farming, particularly sheep farming, it is necessary to increase the survival rate of livestock, which depends on many factors. Animal genotype, feeding and housing conditions, and maternal health affect the viability and health of young animals. To improve the survival rate of livestock, various adaptogens are used as feed additives, injections, or other routes of administration.

 To improve the quality of work, I recommend the following:

  1. In the heading of Table 2, the phrase "BCS was measured on the day of melatonin implant administration during gestation on all properties, and at tail docking and weaning at Franklin River" should be moved to the Material and Methods.
  2. Shorten the title of Table 3, and place the phrase "Data presented for lambs at tail docking (marking) and weaning with data combined for the three sites and between individual sites" in a note to the table. Remove the phrase "Data are expressed as % and number of animals (n)".
  3. Remove the text in brackets in lines 158 and 161.
  4. Shorten the table 4 title.
  5. Line 218, reference to the source

Author Response

Open Review

Quality of English Language

( ) The English could be improved to more clearly express the research.
(x) The English is fine and does not require any improvement.

Yes

Can be improved

Must be improved

Not applicable

Does the introduction provide sufficient background and include all relevant references?

( )

(x)

( )

( )

Is the research design appropriate?

(x)

( )

( )

( )

Are the methods adequately described?

(x)

( )

( )

( )

Are the results clearly presented?

( )

(x)

( )

( )

Are the conclusions supported by the results?

(x)

( )

( )

( )

Comments and Suggestions for Authors

To improve the efficiency of livestock farming, particularly sheep farming, it is necessary to increase the survival rate of livestock, which depends on many factors. Animal genotype, feeding and housing conditions, and maternal health affect the viability and health of young animals. To improve the survival rate of livestock, various adaptogens are used as feed additives, injections, or other routes of administration.

 To improve the quality of work, I recommend the following:

Comment 1: In the heading of Table 2, the phrase "BCS was measured on the day of melatonin implant administration during gestation on all properties, and at tail docking and weaning at Franklin River" should be moved to the Material and Methods.

Response 1: The following has been inserted in line 111 after “Table 2”; “BCS was measured on the day of melatonin implant administration during gestation on all properties, and at tail docking and weaning at Franklin River”

Comment 2: Shorten the title of Table 3, and place the phrase "Data presented for lambs at tail docking (marking) and weaning with data combined for the three sites and between individual sites" in a note to the table. Remove the phrase "Data are expressed as % and number of animals (n)".

Response 2: The following has been placed as a footnote to Table 3:

"Data presented for lambs at tail docking (marking) and weaning with data combined for the three sites and between individual sites"

The following has been deleted from Table 3 title:

"Data are expressed as % and number of animals (n)".

Comment 3: Remove the text in brackets in lines 158 and 161.

Response 3: Fixed

Comment 4: Shorten the table 4 title.

Response 4: We have deleted “Mean” and “a” from line 163

Comment 5: Line 218, reference to the source

Response 5: L220. Added citation. The Haslin et al reference has been given in the reference list
